# Patient-reported outcomes and neurotoxicity markers in patients treated with bispecific LV20.19 CAR T cell therapy

Jennifer M. Knight [1,2✉], Aniko Szabo[3], Igli Arapi[1], Ruizhe Wu[3], Amanda Emmrich[1], Edward Hackett[4], Garrett Sauber[5], Sharon Yim [6], Bryon Johnson[6], Parameswaran Hari [6], Dina Schneider[7], Boro Dropulic[7], Rachel N. Cusatis[8], Steve W. Cole[9], Cecilia J. Hillard [5,10] & Nirav N. Shah [6,10]

## Abstract

**Background** With the rising number of chimeric antigen receptor (CAR) T cell treated patients, it is increasingly important to understand the treatment's impact on patient-reported outcomes (PROs) and, ideally, identify biomarkers of central nervous system (CNS) adverse effects.

**Methods** The purpose of this exploratory study was to assess short-term PROs and serum kynurenine metabolites for associated neurotoxicity among patients treated in an anti-CD20, anti-CD19 (LV20.19) CAR T cell phase I clinical trial (NCT03019055). Fifteen CAR T treated patients from the parent trial provided serum samples and self-report surveys 15 days before and 14, 28, and 90 days after treatment.

**Results** Blood kynurenine concentrations increased over time in patients with evidence of neurotoxicity ($p = 0.004$) and were increased in self-reported depression ($r = 0.52$, $p = 0.002$). Depression improved after CAR T infusion ($p = 0.035$). Elevated 3-hydroxyanthranilic acid (3HAA) concentrations prior to cell infusion were also predictive of neurotoxicity onset ($p = 0.031$), suggesting it is a biomarker of neurotoxicity following CAR T cell therapy.

**Conclusions** Elevated levels of kynurenine pathway metabolites among CAR T cell recipients are associated with depressed mood and neurotoxicity. Findings from this exploratory study are preliminary and warrant validation in a larger cohort.

## Plain language summary

This study examined the impact of chimeric antigen receptor (CAR) T cell therapy—a therapy that gets immune cells to fight cancer by changing them in the lab to find and destroy cancer cells—on blood markers associated with depression, anxiety, pain, fatigue, and poor sleep. Fifteen CAR T cell patients provided blood samples and completed surveys before and three timepoints after treatment. We found that the amount of kynurenine, a normal blood constituent, and related molecules was higher in patients who experienced significant CAR T cell side effects on the brain and in patients reporting more depression. These results identify the excessive elevation of blood constituents related to the mood that may also be associated with depression and brain dysfunction following CAR T. These blood constituents could potentially be used as markers and targeted with interventions to prevent brain dysfunction.

[1] Department of Psychiatry, Medical College of Wisconsin, Milwaukee, WI, USA. [2] Departments of Medicine and Microbiology & Immunology, Medical College of Wisconsin, Milwaukee, WI, USA. [3] Division of Biostatistics, Medical College of Wisconsin, Milwaukee, WI, USA. [4] Medical College of Wisconsin, Milwaukee, WI, USA. [5] Department of Pharmacology and Toxicology and Neuroscience Research Center, Medical College of Wisconsin, Milwaukee, WI, USA. [6] BMT & Cellular Therapy Program, Division of Hematology & Oncology, Medical College of Wisconsin, Milwaukee, WI, USA. [7] Lentigen, a Miltenyi Biotec company, Gaithersburg, MD, USA. [8] Department of Medicine, Division of Hematology & Oncology, Medical College of Wisconsin, Milwaukee, WI, USA. [9] Departments of Psychiatry and Biobehavioral Sciences and Medicine, Division of Hematology-Oncology, David Geffen School of Medicine at UCLA, Los Angeles, CA, USA. [10] These authors contributed equally: Cecilia J. Hillard, Nirav N. Shah. ✉email: jmknight@mcw.edu

The use of chimeric antigen receptor (CAR) T cell therapy is rising rapidly in recent years, increasing from only a few US patients receiving it in 2017, to over 1000 patients in 2019[1]. These numbers are expected to increase near exponentially, replacing other therapeutic options for the treatment of blood cancers, including hematopoietic stem cell transplant (HCT)[2].

Some patients receiving CAR T cell therapies experience central nervous system (CNS) neurotoxicity that presents in the form of delirium, confusion, tremor, seizures, and in rare cases life-threatening cerebral edema[3]. Although the pathophysiology of these neurological effects is not clear, inflammatory cytokines are thought to be significant contributors[3]. While CAR T neurotoxicity itself has not yet been established as a precursor to subsequent changes in mood and other related symptoms, neuroinflammation, in general, is associated with such changes in other populations[4], with the kynurenine pathway as a critical link between inflammation, mood disorders and related symptomatology, and alteration of brain signaling pathways[5–9].

There is evolving evidence that CAR T in general can have more subtle effects on patient-reported outcomes (PROs) as a result of CNS effects. For example, Maziarz et al. demonstrated that patients achieving complete or partial response had sustained PRO improvement at 12 and 18 months[10], while Ruark et al. found that patients receiving CAR T cell therapy are at risk for longer-term PRO impairments, including the development of depression, anxiety, fatigue, sleep disturbances, and pain[11]. Given that these effects can also be associated with neuroinflammation, it has been suggested that CAR T use results in a CNS inflammatory state, which can lead to both reduced quality of life (QOL) and neurotoxicity[3,12].

There are several potential mechanisms by which CAR T mediated therapeutic and adverse effects could influence and/or be affected by PRO measures. First, CAR T treatments are associated with immune activation, inflammation, and cytokine signaling[3,12], all of which are known to affect cognitive and emotional functioning[13], including among cancer patients[14]. Second, CAR T therapy is associated with significant CNS toxicity. Not only do negative emotional and cognitive responses influence immunity and T cell function among cancer patients generally[15], but a recent review also highlighted that CRS and neurotoxicity delayed QOL improvement among CAR T recipients specifically[16]. Finally, these adverse psychosocial responses are predictive of compromised immune reconstitution[17] and adverse outcomes following HCT[18–21]. Therefore, it is critical to understand the interaction between behavioral and biological processes in the setting of CAR T cell therapy and identify predictive characteristics and biomarkers of neurotoxicity that will allow for risk stratification and potentiate early intervention.

The kynurenine pathway is activated by inflammatory cytokines released in the brain and leads to the production of kynurenine from tryptophan (TRP) as a result of upregulation of the enzyme indoleamine 2,3-dioxygenase (IDO) by astrocytes and microglia (brain-resident macrophages). Kynurenine is further metabolized to the neurotoxic metabolites 3-hydroxykynurenine (3HK), 3-hydroxyanthranilic acid (3HAA), and quinolinic acid (QA) and to the neuroprotective metabolite, kynurenic acid (KA) (Fig. 1). The ratio of serum KA/3HK has been used previously as an index of the role of this inflammatory pathway on brain function[22], with lower ratios noted in patients with mood disorders as compared to healthy individuals[23]. Santomasso et al.[24] observed significantly elevated levels of QA in the cerebral spinal fluid (CSF) of CAR T recipients during neurotoxicity compared to pretreatment; however, they did not evaluate other kynurenine pathway metabolites or examine their presence in circulation.

Despite substantial interest in understanding the impact of CAR T cell therapy on PROs and identifying predictive biomarkers[25–28], there are limited data available currently. Biologic indicators such as ferritin, platelet numbers, lactate dehydrogenase (LDH)[29], and fibrinogen[30] have been associated with CAR T neurotoxicity with single targeting CD19 CAR products. Elevated cytokines from cerebral spinal fluid were also associated with neurotoxicity in a cohort of CAR T patients[24]. However, it is unknown whether disruption in the kynurenine pathway—a link between inflammation and the brain – is associated with PROs or predictive of CAR T neurotoxicity.

Here, we describe the PRO trajectory in the early treatment period of CAR T therapy among patients treated in phase 1, *first-in-human*, bispecific, lentiviral, anti-CD20, anti-CD19 (LV20.19) CAR T cell clinical trial[31]. Further, we provide preliminary data that circulating concentrations of kynurenine pathway components are associated with both neurotoxicity and PROs. Blood kynurenine concentrations increased over time in patients with evidence of neurotoxicity and were increased in self-reported depression, with depression improving after CAR T infusion. Elevated baseline 3HAA concentrations prior to cell infusion were also predictive of neurotoxicity onset. These exploratory findings suggest that blood kynurenine pathway metabolites could be used as biomarkers of and potentially targeted to prevent neurotoxicity following CAR T cell therapy.

## Methods

**Study population**. The current study population is derived from a parent study evaluating 22 patients treated with LV20.19 CAR T cells in Phase I/Ib clinical trial (NCT03019055). See Shah et al. for full study details and for original data please contact the corresponding author[31]. No deaths were attributed to LV20.19 CAR T cell therapy. In this study we conducted a PRO and neurotoxicity biomarker sub-study focused on 15 patients with relapsed, refractory B-cell non-Hodgkin's lymphoma or chronic lymphocytic leukemia/small lymphocytic lymphoma all treated at the selected dose from the phase 1 trial ($2.5 \times 10^6$ cells/kg) for homogeneity. Patients were recruited for this sub-study following Institutional Review Board (IRB) approval of a PRO and biomarker amendment (starting with patient 7 of the parent study; IRB PRO00037171). No blinding was used in the open-label study. The sub-study cohort included patients who provided PRO data and survived through a day 28 assessment. Administration of CAR T cells occurred through the Medical College of Wisconsin (MCW) HCT Program under an FDA IND 17518. All participants provided written informed consent and all procedures were approved in advance by the MCW IRB.

**Patient-reported outcomes (PROs)**. Study subjects completed a series of self-report surveys at the following timepoints: baseline/apheresis (Day −15 with respect to Day 0 being day of CAR T cell infusion), Day +14 (D14), D28, and D90 post-infusion. To provide a dimensional assessment of depressive symptoms, the 20-item General Depression subscale from the Inventory of Depression and Anxiety Symptoms (IDAS) was used[32]. Questionnaires administered at timepoints listed above included the Inventory of Depression and Anxiety Symptoms (IDAS)[32], Fatigue Symptom Inventory (FSI; fatigue)[33], Pittsburgh Sleep Quality Index (PSQI; sleep)[34], and Brief Pain Inventory (BPI; pain)[35]. IDAS scoring ranges from 20 to 100 for depression and anxiety, with a mean depression subscale score for a community-dwelling adult of 44.99 (SD = 14.75)[36]. Anxiety was assessed as a sum of two IDAS subscales scores—panic disorder (healthy population mean = 12.58, SD = 5.26) and traumatic intrusions (healthy population mean = 7.60, SD = 4.20)[36]. FSI was evaluated based on fatigue intensity (FSII) and interference (FSIF) (both scales ranged 0–10 with >3 being clinically significant; FSII) and the

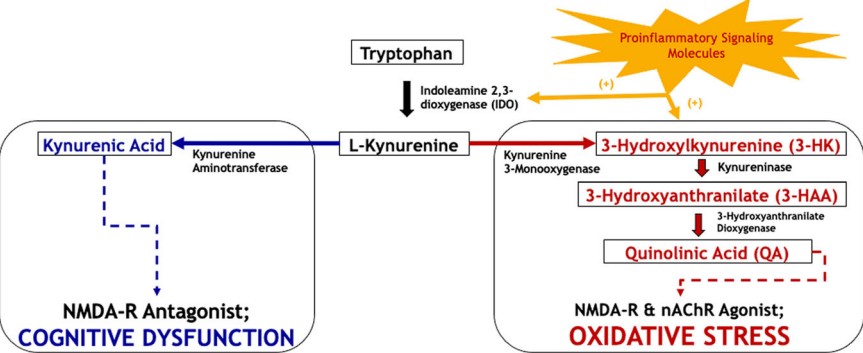

**Fig. 1 Kynurenine metabolic pathway.** 3-HK 3-Hydroxykynurenine, 3-HAA 3-Hydroxyanthranilic acid, QA Quinolinic acid, NMDA-R N-methyl-D-aspartate receptor, nAChR Nicotinic acetylcholine receptor.

number of days within the past week patients felt fatigued (0–7; FSID)[37]. Sleep scores >5 are considered disturbed sleep as adjusted for cancer populations[34,38]. Brief Pain Index scores were subdivided and assessed based on pain intensity (0–40; BPII) and pain interference in daily activities (0–10; BPIF)[39]. Any patients endorsing thoughts of suicidality or self-harm per the IDAS were contacted by Dr. Knight and offered appropriate follow-up care.

**Neurotoxicity.** Study participants were evaluated for neurotoxicity using the National Cancer Institute Common Terminology Criteria for Adverse Events (CTCAE) v5[40] and CRS was graded using the Lee et al. CRS grading system[41]. Neurotoxicity was retrospectively regraded utilizing the updated American Society for Transplantation and Cellular Therapy (ASTCT) system[42] with no change compared to the CTCAE (see Supplemental Appendix Table 3 in Shah et al.)[31].

**Measurement of tryptophan and metabolites.** Blood was obtained from subjects at D−15 (considered baseline values), D14, D28, and D90 and serum were harvested and frozen at −80 °C until assay. Serum samples were thawed and 200 μL was transferred to 1 mL of acetonitrile containing 100 ng each of deuterated tryptophan ($[^2H_5]$-TRP), kynurenic acid ($[^2H_5]$-KA), quinolinic acid ($[^2H_3]$-QA), and 3-hydroxyanthralinic acid ($[^2H_2]$-3HAA). Samples were vortexed and then placed into an ultrasonication bath for 2 min. Protein precipitation was facilitated by transferring samples to −20 °C freezer for thirty minutes, followed by centrifugation at 14,000 rcf for 10 min at 4 °C. The resulting supernatants were dried down at room temperature under a gentle nitrogen stream. After resuspension in 100 μL acetonitrile, samples were redried, resuspended in 100 μL acidified mobile phase, transferred to autosampler vials, and stored at 4 °C until analysis.

The concentrations of TRP and metabolites (KA, QA, 3HAA, kynurenine, and 3HK) were quantified in 5 μl of the mobile phase extract using stable isotope-dilution liquid chromatography/mass spectrometry of the daughter ions (LC-MS-MS). Reversed-phase high-performance liquid chromatography (HPLC) using a Kromasil C18 column (150 × 2.1 mm, 5 μm particle size) and a mobile phase of 0.1% formic acid, 2 mM Ammonium acetate in ddH2O (phase A) and acetonitrile (phase B) was used to separate the metabolites. Analytes were separated at a flow rate of 0.2 mL/min with a step gradient of 0% B for 0–1.5 min, 75% B from 3.5–5 min, 85% B from 8–9 min, and 0% B at 11–14 min. Mass spectrometric detection was performed using a tandem quadrupole mass spectrometer (Agilent 6460) equipped with an electrospray ionization (ESI) source. Detection was performed in the positive mode. The quantification was performed using the

multiple reaction monitoring modes with the following $m/z$ transitions monitored: 205.1/146 for TRP, 209.1/94 for kynurenine, 225.1/110.1 for 3HK, 190/144 for KA, 168/78.1 for QA, 154/80.1 for 3HAA, 210.1/150.1 for $[^2H_5]$-TRP, 195.1/149.1 for $[^2H_5]$-KA, 171/81.1 for $[^2H_3]$-QA, and 156/82 for $[^2H_2]$-3HAA. The assay utilized was designed to analyze all kynurenine metabolites in a single serum sample and was therefore not optimized to measure QA in particular; subsequently, there were only values for 6 participants at baseline and 4 at D90.

Standard curves were generated for all analytes in a range of 10–5000 pg/μL, and internal standards $[^2H_5]$-TRP, $[^2H_5]$-KA, $[^2H_3]$-QA, and $[^2H_2]$-3HAA (each at 1000 pg/μL). Concentrations of the analytes in the samples were determined from standard curves of the area ratios (standard/analyte) versus the concentration ratios (standard/analyte); the corresponding deuterated compounds were used as the standard for the unlabeled analytes; $[^2H_5]$-TRP was used as the standard for kynurenine and 3HK in addition to TRP.

**Serum cytokine assessment.** Peripheral blood serum samples were collected from each subject at several timepoints during the first 28 days following LV20.19 CAR cell infusion. Samples were immediately frozen at −80 °C and then shipped on dry ice to Eve Technologies (Calgary, Alberta Canada), where they were analyzed using a 65-plex human cytokine and chemokine panel (HD65). Each sample was analyzed in duplicate, and the average values determined. Peak levels of 10 cytokines in the panel were found to be significantly elevated in the serum of patients who experienced neurotoxicity[31].

**Statistical analysis**

*Descriptive analyses.* Categorical variables were summarized using counts and percentages, while continuous variables with median and range or interquartile range, as indicated. Between-group comparisons were performed using the chi-square test and Wilcoxon's rank-sum test, respectively. Student's *t*-tests were used to evaluate baseline differences in PROs between patients who completed visit 4 on D90 vs those that did not.

*Primary analysis.* A mixed-effects longitudinal model was fitted for all eligible patients to evaluate the impact of time on all PRO variables over 4 encounters through 90 days of post-study intervention. The measurement time was a fixed categorical predictor, and a subject-specific random intercept was used to account for within-patient dependence. This approach allows for an unbalanced design in which not every subject is measured at every timepoint. A mixed-effects logistic regression model was used to evaluate the changes in sleep quality over time.

*Secondary analyses.* Neurotoxicity was evaluated as a categorical variable with levels 0–1 and 3–4, as no subject in our study had grade 2 toxicity. Kynurenine metabolite values were background corrected. Zero values were considered to be below the limit of detection and replaced by half of the smallest non-negative value. The measurements were log-transformed to stabilize the variances and allow modeling of the expected multiplicative effects. The effect of time and neurotoxicity grade were evaluated separately for each metabolite using mixed-effects linear models with a random subject intercept, and fixed effects of time, neurotoxicity grade, and their interaction. Spearman's rank correlation was used to evaluate the association between changes in kynurenine and concurrent changes in depression scores, and to compare kynurenine metabolites at baseline/D−15, D14, and D28 with peak concentrations of inflammatory cytokines by D28.

**Reporting summary**. Further information on research design is available in the Nature Research Reporting Summary linked to this article.

## Results

**Patient characteristics**. Data were available for 15, 13, 14, and 13 patients at baseline (15 days prior to cell infusion), D14, D28, and D90, respectively. Patient and disease characteristics overall and by neurotoxicity grade are described in Table 1. The symptomatology, timing, and duration of neurotoxicity symptoms are included in Supplemental Table 1. The majority of patients receiving CAR T cell therapy were male (93%), sustained complete response (80%), and experienced CRS (80%). Forty percent of study participants completed college and over half of the patients earned less than $55,000 annually. The only significant difference between neurotoxicity groups was the presence of higher baseline LDH among those who experienced grade 3/4 vs. 0/1 neurotoxicity (1,283 vs. 196; $p = 0.004$).

**PROs over time and neurotoxicity**. Depression scores using the IDAS questionnaire changed significantly over time such that they were higher at D14 and lower at D90 (39.4 vs 33.4, $p = 0.035$), indicating similarly elevated depression scores through D28 that subsequently decreased by D90. The remaining PRO variables did not demonstrate significant differences over time (Table 2; Fig. 2). Over half of the patients endorsed that they maintained good sleep quality throughout the study duration (Table 3). There was no correlation between any of the PRO variables and neurotoxicity grade.

**Kynurenine metabolites and neurotoxicity**. Kynurenine metabolite assessments indicate that TRP, kynurenine, and KA concentrations significantly changed over time ($p = 0.014$, 0.004, and 0.004, respectively). Kynurenine concentrations changed most dramatically in patients with Grade 3/4 neurotoxicity, with concentrations rising at D14 and D28 followed by a decrease by D90 (Fig. 3). Baseline concentrations of 3HAA, KA, and 3HK are elevated among patients who eventually exhibited neurotoxicity; this trend was not observed in any of the other metabolites (Fig. 3). When kynurenine metabolites were compared by neurotoxicity grade over the study period, 3HAA and 3HK were elevated in patients with neurotoxicity grades of 3/4 compared to those with grades of 0/1 ($p = 0.031$ and $p = 0.10$, respectively). Although kynurenine, KA, and QA concentrations were also elevated in patients with grade 3/4 neurotoxicity compared to those with grade 0/1 neurotoxicity, the differences were not significant (Fig. 3).

**Depression and kynurenine**. As previous studies have shown a significant, positive association of circulating kynurenine with depression[5], we evaluated the association between the change in reported depression and the change in circulating kynurenine in this sample. The change in kynurenine concentrations between timepoints was significantly correlated to the change in depression scores between the same timepoints ($p = 0.002$; Fig. 4; Table 4).

**Cytokines and kynurenine metabolites**. In general, TRP metabolites trended to be negatively correlated over time with the cytokines assessed while the downstream neurotoxic metabolite QA trended to be positively correlated with the same cytokines (Fig. 5). In particular, when evaluating specific cytokines associated with CAR T-associated neurotoxicity (B-Cell Attracting cytokine 1, BCA-1; Fractalkine, Granulocyte Colony Stimulating Factor, G-CSF; Chemokine (C-C motif) ligand 1, I-309; Interferon gamma, IFNg; Interleukin 6, IL-6, and 8, IL-8; Interferon gamma-induced protein 10, IP-10; Monocyte chemotactic protein 2, MCP-2; Tumor Necrosis Factor alpha, TNFa), IL-6, IL-8, and TNFa were negatively correlated with tryptophan and increasingly positively correlated with kynurenine and neurotoxic metabolites 3HK, 3HAA, and QA while showing little-to-no positive correlation with the neuroprotective metabolite KA (Fig. 5).

## Discussion

Here we present hypothesis-generating correlative findings from a single-arm, unblinded, open-label trial of bispecific LV20.19 CAR T cell therapy upon which to guide future investigation. These preliminary data suggest that CAR T therapy does not impair PROs and may in fact result in improved mood in the early months following cell infusion, adding to recent literature about long-term PROs in CAR T survivors[10,11]. On the other hand, the kynurenine pathway metabolites demonstrate significant relationships with clinical evidence of neurotoxicity in this cohort of CAR T patients, despite only 3 individuals developing neurotoxicity. In particular, individuals with worse neurotoxicity scores exhibited elevated 3HAA and 3HK concentrations throughout the study period while kynurenine increased significantly over time in patients with higher-grade neurotoxicities. Increases in kynurenine concentrations were significantly associated with increased self-reported depression, consistent with other findings regarding kynurenine and depression[5]. Cytokines previously affiliated with CAR T neurotoxicity were also associated with neurotoxic kynurenine pathway metabolites in a manner suggestive of its inflammatory-induced stimulation. Together, these preliminary data suggest that enhanced flux of metabolites through the kynurenine pathway occurs during the depression and could reflect neuroinflammation. Our preliminary finding that circulating concentrations of 3HAA are higher prior to cell therapy in those who develop neurotoxicity suggests that it could serve as a component of a biomarker risk panel for serious CNS complications and hints that baseline increases in the kynurenine pathway could contribute to vulnerability to CNS toxicity.

Given the significant morbidity and mortality associated with CAR T neurotoxicity[3], it is critical to identify candidate predictive biomarkers for risk stratification[24,25,27]. Further, given the pro-inflammatory effects of CAR T therapy, it is important to identify patients at risk for longer-term survivorship issues to better facilitate early intervention and symptom management. Patients with subtle or clinically undetectable neuroinflammation prior to CAR T therapy may be at particular risk to develop clinically relevant symptoms of CNS toxicity following CAR T therapy-induced CRS based on other evidence regarding evolving neuropathology[43–45].

**Table 1 Baseline patient demographics.**

| Baseline characteristics | Overall, N = 15[a] | NTX Grade 0–1, N = 12[a] | NTX Grade 3–4, N = 3[a] | p-value |
|---|---|---|---|---|
| Age | 61 (38–72) | 60 (38–69) | 66 (47–72) | 0.61 |
| Male sex | 14 (93%) | 12 (100%) | 2 (67%) | 0.20 |
| Level of education | | | | 0.76 |
| High school | 2 (13%) | 2 (17%) | 0 (0%) | |
| Trade school | 4 (27%) | 4 (33%) | 0 (0%) | |
| Some college | 3 (20%) | 2 (17%) | 1 (33%) | |
| College graduate | 3 (20%) | 2 (17%) | 1 (33%) | |
| Post graduate degree | 3 (20%) | 2 (17%) | 1 (33%) | |
| Income | | | | 0.94 |
| $10,001–25,000 | 2 (13%) | 1 (8.3%) | 1 (33%) | |
| $25,001–40,000 | 3 (20%) | 2 (17%) | 1 (33%) | |
| $40,001–55,000 | 3 (20%) | 2 (17%) | 1 (33%) | |
| $55,001–70,000 | 3 (20%) | 3 (25%) | 0 (0%) | |
| $85,001–100,000 | 2 (13%) | 2 (17%) | 0 (0%) | |
| >$100,000 | 2 (13%) | 2 (17%) | 0 (0%) | |
| Histology | | | | >0.99 |
| CLL | 2 (13%) | 2 (17%) | 0 (0%) | |
| DLBCL | 9 (60%) | 7 (58%) | 2 (67%) | |
| FL | 1 (6.7%) | 1 (8.3%) | 0 (0%) | |
| MCL | 3 (20%) | 2 (17%) | 1 (33%) | |
| Baseline LDH | 203 (121–2,074) | 196 (121–490) | 1283 (590–2074) | 0.004 |
| Lines of prior therapy | 4 (3–6) | 4 (2–7) | 4 (3–11) | 0.55 |
| Prior allogeneic HCT | 1 (6.7%) | 1 (8.3%) | 0 (0%) | >0.99 |
| Prior autologous HCT | 5 (6.7%) | 5 (42%) | 0 (0%) | 0.51 |
| Clinical response on day 28 | | | | 0.23 |
| CR | 12 (80%) | 10 (83%) | 2 (67%) | |
| PD | 1 (6.7%) | 0 (0%) | 1 (33%) | |
| PR | 2 (13%) | 2 (17%) | 0 (0%) | |
| CRS (Yes) | 12 (80%) | 9 (75%) | 3 (100%) | >0.99 |
| Day to CRS | 2.5 (0.0–10.0) | 6.0 (1.0–10.0) | 1.0 (0.0–3.0) | 0.16 |
| Max Grade CRS | | | | 0.32 |
| 0 | 3 (20%) | 3 (25%) | 0 (0%) | |
| 1 | 7 (47%) | 6 (50%) | 1 (33%) | |
| 2 | 4 (27%) | 3 (25%) | 1 (33%) | |
| 4 | 1 (6.7%) | 0 (0%) | 1 (33%) | |
| NTX (Yes/no) | 5 (33%) | 2 (17%) | 3 (100%) | |
| Max NTX Grade | | | | |
| 0 | 10 (67%) | 10 (83%) | 0 (0%) | |
| 1 | 2 (13%) | 2 (17%) | 0 (0%) | |
| 3 | 2 (13%) | 0 (0%) | 2 (67%) | |
| 4 | 1 (6.7%) | 0 (0%) | 1 (33%) | |
| Days to NTX | 6.0 (0.0–9.0) | 7.5 (6.0–9.0) | 1.0 (0.0–6.0) | |

CLL chronic lymphocytic leukemia, CR complete response, CRS cytokine release syndrome, DLBCL diffuse large B-cell lymphoma, FL follicular lymphoma, HCT hematopoietic cell transplantation, LDH lactate dehydrogenase, MCL mantle cell lymphoma, NTX neurotoxicity, PD progressive disease, PR partial response.
[a]N (%), median (range).

**Table 2 Patient-reported outcome (PRO) means over time[a].**

| PROs | Baseline, N = 15 | | Day 14, N = 13 | | Day 28, N = 14 | | Day 90, N = 13 | | P-value |
|---|---|---|---|---|---|---|---|---|---|
| | Mean | IQR | Mean | IQR | Mean | IQR | Mean | IQR | |
| Anxiety | 16 | 14, 18 | 15 | 12, 22 | 20 | 13, 21 | 13 | 12, 18 | 0.95 |
| Depression | 37 | 30, 42 | 39 | 26, 42 | 37 | 30, 43 | 31 | 26, 35 | 0.035 |
| BPIF | 1.3 | 0.4, 2.1 | 0.0 | 0.0, 1.7 | 0.29 | 0.0, 4.3 | 0.57 | 0, 1.0 | 0.067 |
| BPII | 2.0 | 1.0, 2.9 | 0.8 | 0.1, 3.1 | 1.8 | 0.5, 2.8 | 2.0 | 0.0, 3.3 | 0.67 |
| FSID | 7.0 | 5.5, 9.0 | 9.0 | 3.0, 13.5 | 8.0 | 5.0, 12.0 | 7.0 | 4.5, 11.5 | 0.88 |
| FSIF | 1.57 | 0.9, 2.9 | 1.1 | 0.2, 3.6 | 1.4 | 0.0, 4.0 | 1.1 | 0.1, 1.6 | 0.17 |
| FSII | 3.5 | 2.5, 4.1 | 3.9 | 2.0, 4.8 | 3.3 | 2.5, 4.3 | 3.2 | 2.1, 4.4 | 0.42 |
| PSQI | 5.0 | 2.5, 6.5 | 3.0 | 3.0, 5.0 | 6.0 | 3.2, 7.0 | 4.0 | 2.0, 6.0 | 0.92 |

BPIF Brief Pain Inventory Fatigue, BPII Brief Pain Inventory Intensity, FSID Fatigue Symptom Inventory Duration, FSIF Fatigue Symptom Inventory Interference, FSII Fatigue Symptom Inventory Intensity, PSQI Pittsburgh Sleep Quality Index
[a]Mixed effects model with random ID effect utilized to derive p-values.

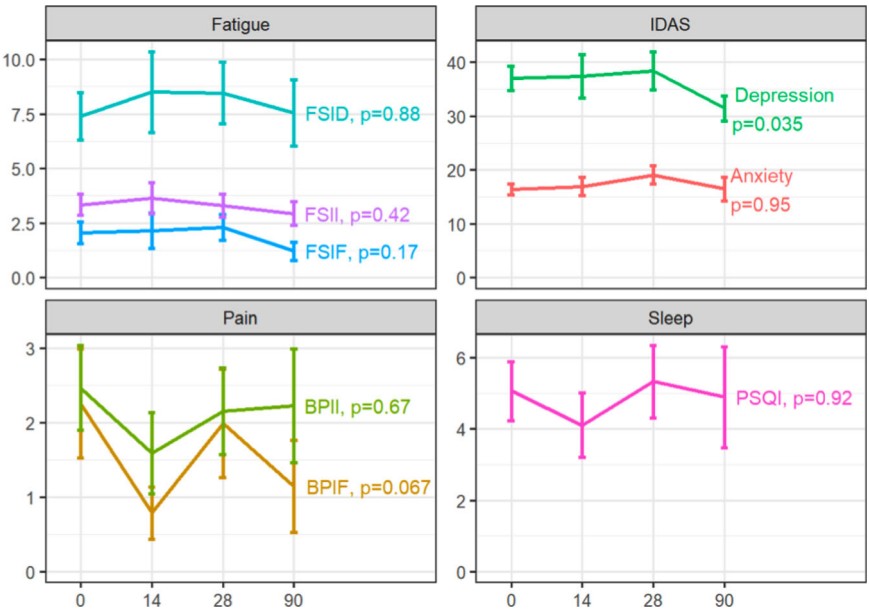

**Fig. 2 Patient-reported outcome (PRO) mean changes over time.** BL baseline, BPIF Brief Pain Inventory Fatigue, BPII Brief Pain Inventory Intensity, FSID Fatigue Symptom Inventory Duration, FSIF Fatigue Symptom Inventory Interference, FSII Fatigue Symptom Inventory Intensity, IDAS Inventory of Depression and Anxiety Symptoms, PSQI Pittsburgh Sleep Quality Index. Pain, fatigue, depression, and anxiety were analyzed via a mixed effect longitudinal model fitted for all eligible patients ($n = 15$) and sleep was analyzed via a mixed effect logistic regression model ($p = 0.035$ for depression; remaining variables did not demonstrate significant differences over time). Error bars indicate standard error of the mean.

| Table 3 Patient sleep quality scores. | | |
|---|---|---|
| **Timepoint** | **Sleep quality** | **Number of patients $N = 15$ (%)** |
| Baseline ($N = 15$) | Good sleep | 9 (60%) |
| | Poor sleep | 6 (40%) |
| Day 14 ($N = 9$) | Good sleep | 4 (44%) |
| | Poor sleep | 5 (56%) |
| Day 28 ($N = 12$) | Good sleep | 7 (58%) |
| | Poor sleep | 5 (42%) |
| Day 90 ($N = 10$) | Good sleep | 5 (50%) |
| | Poor sleep | 5 (50%) |

In the current exploratory study, we identify 3HAA and 3HK as being associated with neurotoxicity following CAR T cell therapy, particularly within the first two weeks following infusion. Together, these metabolites represent potential early indicators of neurotoxicity that could be utilized in a biomarker panel for neurotoxicity risk. In contrast to prior research identifying QA elevations in CSF during neurotoxicity[24], our data did not demonstrate significant associations between QA and neurotoxicity, most likely due to the small number of patients in whom the current protocol was able to detect circulating levels and the small number of patients who developed neurotoxicity. Alternatively, blood perturbations of cytokine and kynurenine pathway metabolites may not be reflective of cerebral spinal fluid/central nervous system changes. The current study aimed to assess the kynurenine pathway in sum; future work may consider using assays optimized to measure QA specifically given its association with neurotoxicity.

It is worth noting that overall this patient cohort did not exhibit large reductions in PROs or signs of neurotoxicity during the 90 days following treatment. Depression and anxiety scores were substantially below cutoff predictive scores for depressive and anxiety disorders (55.5 and 50.5, respectively)[32]. Further, there were lower occurrences of neurotoxicity in this cohort of patients receiving LV20.19 CAR T cells (5 cases or 33%) than

seen with single targeted CD19 CARs[31,46,47]. Despite a low incidence of clinically observable CNS toxicity, there were significant associations of neurotoxicity scores with the kynurenine pathway metabolites, though this will need to be validated in larger cohorts with a higher incidence of neurotoxicity.

Interestingly, there is also a near significant interaction between time and neurotoxicity grade in the variance of KA concentrations, with elevations in KA occurring early after treatment in those with neurotoxicity. As KA is neuroprotective[48], it is possible this reflects a compensatory anti-inflammatory mechanism and could represent a potential intervention target. KA antagonizes N-methyl-D-aspartate (NMDA) and alpha$_7$-nicotinic acetylcholine receptor[48]. Memantine, an approved drug for use in dementia, similarly antagonizes NMDA receptors and has been used in promising studies to prevent neurotoxicity in the setting of whole-brain radiation[49]. Likewise, kynurenine monooxygenase (KMO), which converts kynurenine into 3HK, is a possible target for pharmacologic inhibition, as this would be expected to decrease toxic metabolites and increase KA. KMO inhibitors are currently under development[50] and represent another potential future avenue of neuroprotection. Future research is needed to delineate whether these and other pharmacologic interventions could be helpful in mitigating CAR T neurotoxicity in both the short and long terms.

Our preliminary data are suggestive of a dynamic kynurenine pathway given the temporal relationship between pro-inflammatory cytokines and kynurenine metabolites. In particular, circulating concentrations of TRP, the substrate for IDO, decreases as the pro-inflammatory molecules increase, whereas QA, the metabolite at the end of the pathway, increases with the cytokines (Fig. 5). These exploratory data lend further support to the hypothesis that pro-inflammatory cytokines activate the kynurenine pathway in CAR T recipients through IDO upregulation, the rate-limiting step in the metabolism of TRP to kynurenine[50].

The IDO pathway is increasingly being recognized as a potential therapeutic target in cancer immunotherapy[51]. The depletion of TRP and increase in kynurenine exert important

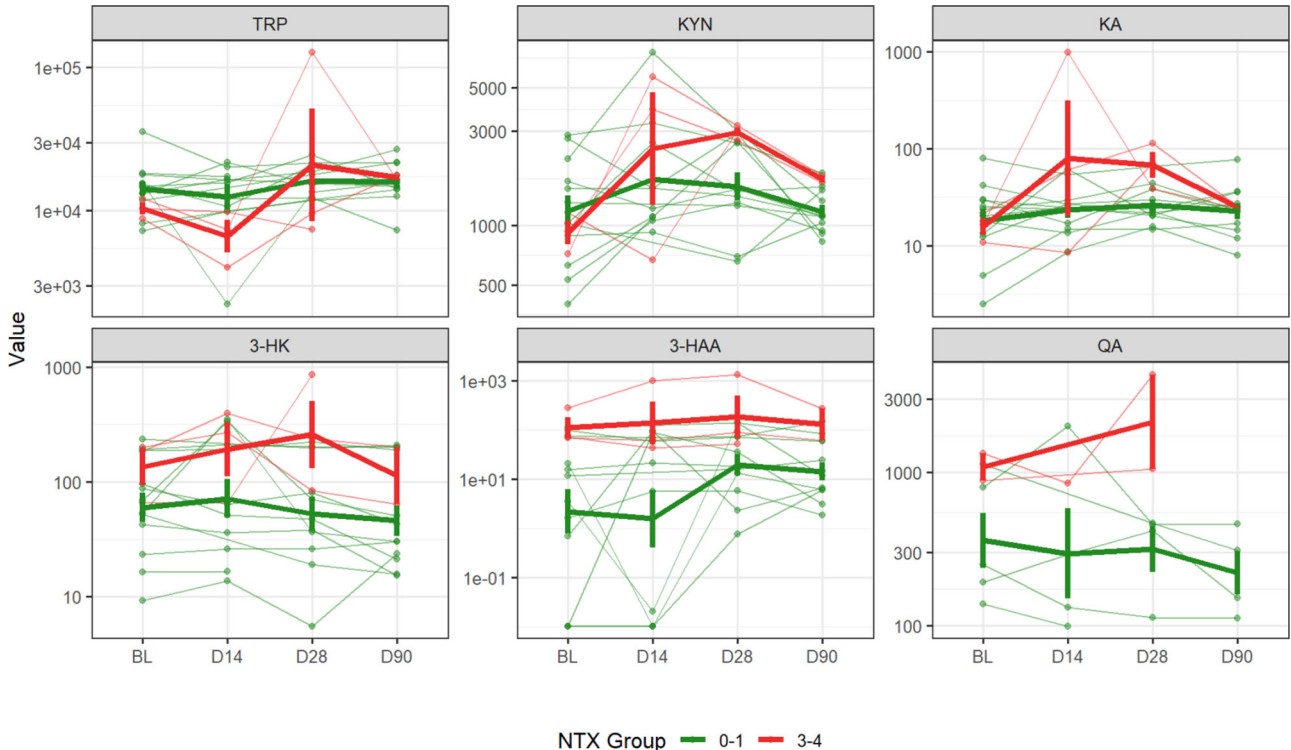

**Fig. 3 Kynurenine metabolites over time compared between low neurotoxicity (grades 0–1) and high neurotoxicity (grades 3–4) groups.** BL baseline, NTX Neurotoxicity, TRP Tryptophan, KYN Kynurenine, KA Kynurenic acid, 3-HK 3-Hydroxykynurenine, 3-HAA 3-Hydroxyanthranilic acid, QA Quinolinic acid. Observations from each subject ($n = 15$) relate to thin lines. Thick lines relate to means with error bars indicating standard error. Linear mixed-effects model showed 3HAA and 3HK were elevated in patients with neurotoxicity grades over the study period ($p = 0.031$ and $p = 0.10$, respectively).

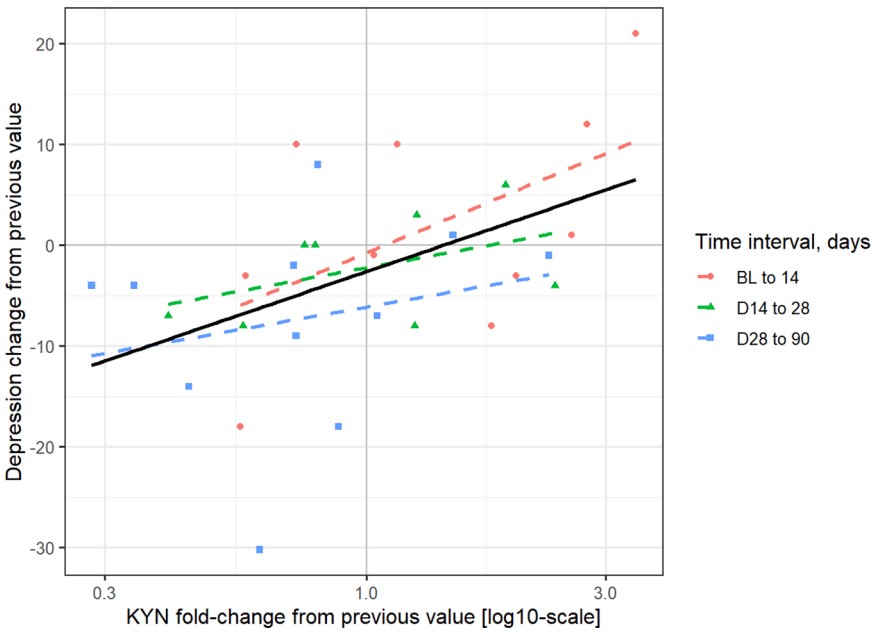

**Fig. 4 Spearman ranked correlation between kynurenine and depression fold-changes.** BL baseline, KYN Kynurenine. Kynurenine concentration changes over time between timepoints was significantly correlated to change in depression scores between the same timepoints ($n = 15$, $p = 0.002$).

immunosuppressive functions by activating T regulatory cells and myeloid-derived suppressor cells while suppressing the functions of effector T and natural killer cells[51]. Inhibition of IDO1 in mice enhances CAR T cell immunotherapy[52], thus avoidance of entering into the neurotoxic kynurenine pathway arm may be an important variable contributing to the success of CAR T cell therapy. Depression is associated with worse medical outcomes in other hematologic malignancies and transplantation populations[53–56], though this has yet to be investigated long-itudinally among CAR T recipients. The strong association between depression and kynurenine in the current study is not only consistent with prior literature[5] but provides a conceptual biologic framework for how the neuroinflammatory condition of depression could contribute to worse CAR T outcomes.

Reciprocally, CAR T-associated neurotoxicity may contribute to depression in the survivorship period among CAR T recipients through CNS perturbation. Additional research is needed to further delineate these effects given the existing biological underpinnings supporting this bidirectional relationship and its relevance to CAR T outcomes in particular.

The study's findings are limited in several respects. First, the sample size from this correlative study is small and under-powered, thus limiting the generalizability of findings. However, we report findings from a first-in-human bispecific LV20.19 CAR. As it has been speculated that CD19 mediated neurotoxicity may be due to off-target toxicity related to the expression of CD19 on brain mural cells[57], our lower rates of toxicity may be partially related to the CD20 engagement of tumor. With the recent significant growth in the utilization of dual targeting CARs, these data will be valuable to future studies. Additionally, given the highly specialized and novel nature of CAR T cell therapy, small sample size does not preclude the impact of the current findings, as other CAR T studies of similar magnitude have demonstrated[11,31,58,59]. Further, the current data is consistent with prior CAR T studies as evidenced by higher baseline LDH among individuals who went on to develop more severe

neurotoxicity[29]. Second, this study had a relatively low incidence of neurotoxicity. While this could make it more difficult to ascertain any significant biomarker predictors of neurotoxicity, the current data are supportive of such a relationship even in the setting of relatively few cases, highlighting its potential strength. Third, the current study examined only a limited number of candidate neurotoxicity biomarkers; future studies should continue to explore these mechanisms in assessing other potential biomarkers of neurotoxicity, including but not limited to neurofilament light[60] and/or glial fibrillary acidic protein[61], as well as their differentiation from CRS pathology. Fourth, it is possible that patient PROs vary as a function of disease-stage knowledge; while this sequential timing was not ascertained in the current study, it will be important to collect for future research. Finally, there was a minimal depression in this study, little disruption, and minimal variability in the other PROs assessed. This should be encouraging for patients in the acute CAR T setting. However, for research purposes this poses a challenge in understanding these early potential relationships. The lack of variability in data may obscure relationships that could be important prognosticators for later outcomes.

Future studies are needed to assess both early and late PROs in larger cohorts. It will be important to assess PROs in patients receiving other CAR T formulations as well. Given the neurotoxic effects of high levels of inflammation[62], it is critical to consider the increased susceptibility of CAR T patients to poor QOL specifically among those who have experienced CAR T-associated neurotoxicity.

Information gathered from this exploratory study provides important and novel insights regarding the impact of CAR T cell therapy on early patient PROs as well as preliminary mechanistic information regarding the relationships between the kynurenine pathway, PROs, and CNS toxicity. These findings help elucidate these metabolites as potential biomarkers for the occurrence of

**Table 4 Spearman ranked correlation between kynurenine fold-change from previous timepoint and change in depression from previous timepoint.**

| Time change | Estimate | P-value |
|---|---|---|
| Baseline to 14 | 0.62 | 0.05 |
| 14 to 28 | 0.48 | 0.227 |
| 28 to 90 | 0.34 | 0.304 |
| Pooled | 0.52 | 0.002 |

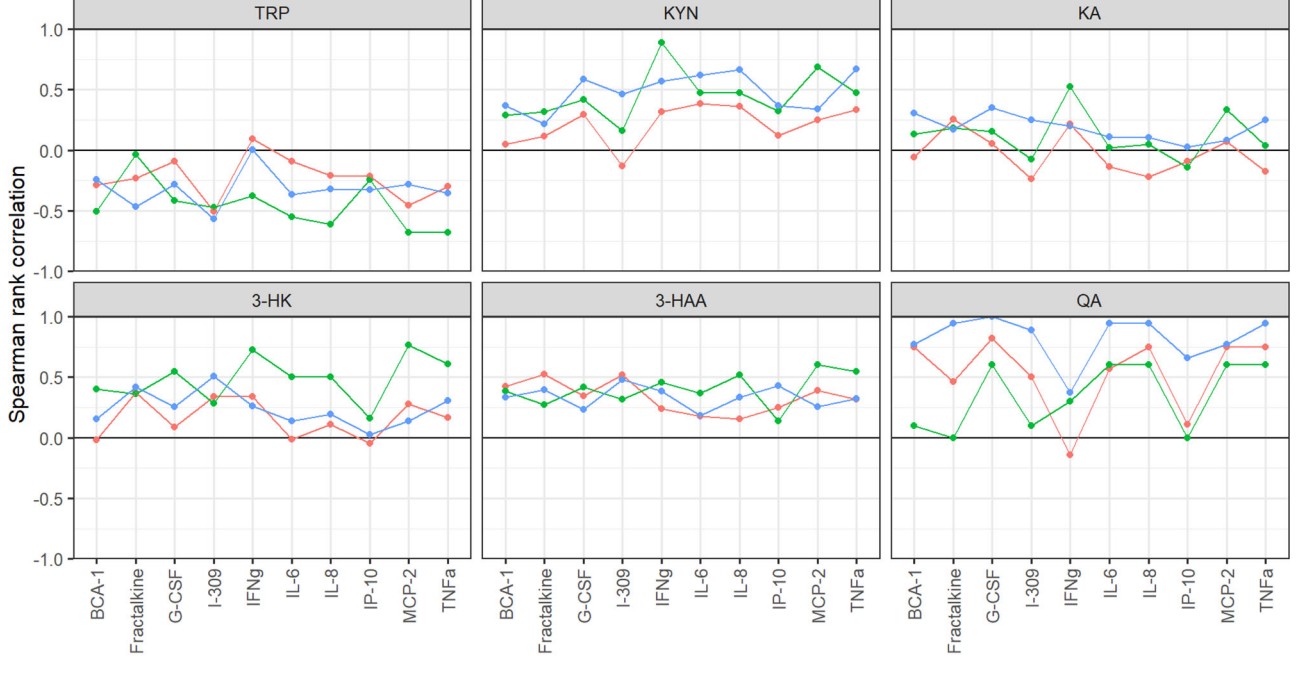

**Fig. 5 Spearman ranked correlations between peak cytokine concentrations and tryptophan metabolites measured across baseline, D14, and D28.** BL baseline, TRP Tryptophan, KYN Kynurenine, KA Kynurenic acid, 3-HK 3-Hydroxykynurenine, 3-HAA 3-Hydroxyanthranilic acid, QA Quinolinic acid. Spearman ranked correlations were done utilizing all available participants ($n = 15$).

CNS toxicity and PROs in patients undergoing CAR T therapy and warrant validation in a larger cohort.

## Data availability

Source data for the figures are available as Supplementary Data 1. Reasonable requests for the individual deidentified participant data collected during the trial should be directed to the corresponding author. All requests will be reviewed by the Medical College of Wisconsin and Lentigen Technology to verify whether the request is subject to any intellectual property obligations or compromises participant confidentiality. Data will be made available for 24 months after publication to those researchers that provide a methodologically sound proposal to achieve aims in the approved proposal. The study protocol will be made available upon similar request.

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

## Acknowledgements

This work was funded in part by the National Center for Advancing Translational Sciences, National Institutes of Health (NIH), through Grant Numbers UL1TR001436 and KL2TR001438; Medical College of Wisconsin Department of Psychiatry and Cancer Center; the Kubly Fund for Depression Research; and the Laura Gralton Philanthropic Fund.

## Author contributions

Designed research: J.M.K., C.J.H., N.S., A.S., and P.H. Performed research: J.M.K., N.S., A.S., G.S., B.J., P.H., and A.E. Contributed vital new reagents or analytical tools: A.S., D.S., and B.D. Collected Data: J.M.K., N.S., G.S., S.Y., B.J., and A.E. Analyzed and interpreted data: J.M.K., C.J.H., A.S., R.W., G.S., E.H., D.S., B.D., and P.H. Performed statistical analysis: A.S., I.A., and R.W. Manuscript writing: J.M.K., C.J.H., A.S., R.W., N.S., E.H., D.S., B.D., R.N.C., S.W.C., B.J., A.E., and P.H.

## Competing interests

The authors declare the following competing interests. P.H. reports receiving honoraria from Incyte, BMS, Legend, Jannsen, Takeda, Amgen, Karyopharm, GSK, Pfizer. C.J.H. is a member of the Scientific Advisory Boards of Phytecs, Inc, and has equity in Formulate Biosciences. B.J. reports receiving research support and honoraria and travel support from Miltenyi Biotec. D.S. and B.D. are authors on a patent for the 20.19 CAR. N.N.S. reports receiving honoraria and/or travel support from Incyte, Celgene, Lily, and Miltenyi Biotec; serving on scientific advisory boards for Lily, Kite, Celgene, Legend, Epizyme, Seattle Genetics, and TG therapeutics; equity ownership in Exelixis, Geron; receiving institutional research support for clinical trials from Miltenyi Biotec. The remaining authors declare no competing interests.
