## [Peer Review File · Communications Medicine]

Reviewers' comments:

Reviewer #1 (Remarks to the Author):

This study by Knight et al. is a sub-study of a CD19/20 bispecific CAR-T early phase clinical trial (Shah et al, NCT03019055), in which patient reported outcomes (PROs) focused on symptoms of pain, sleep, fatigue, depression and anxiety are reported and correlated with toxicity and serum biomarkers. The original contributions the authors report are that the tryptophan metabolite kynurenine blood concentrations increase over time in patients with neurotoxicity, and were correlated with depression PROs. Another tryptophan metabolite, 3-HAA, seemed to be predictive of neurotoxicity onset.

There have been several recent reports on PROs after CAR-T, as the authors mention in the discussion (Maziarz, Blood Advances 2020; Kersten, Value in Health 2020; Ruark, BBMT 2020). These studies involved CD19 CAR-T (as opposed to the CD19/20 bispecific CAR-T in this current manuscript), but it can be presumed, based on similarity in the populations, that the difference in intervention (CD19 vs CD19/20 targeting) does not significantly alter the PROs. However, each of these studies used different validated PRO questionnaires to gather the data, which makes it difficult to compare. However, these studies in aggregate support that QoL is improved in the months after CAR-T therapy compared to prior to CAR-T, and this current manuscript simply adds to this evidence, with decreased depression scores after CAR-T. However, the sample size is small (n =15) so it is difficult to make any major claims about PROs.

The main novelty of the study is in looking at tryptophan metabolites (KA, QA, 3HAA, kynurenine, and 3HK). QA has been reported as associated with neurotoxicity (Santomasso, Cancer Discovery 2018), but I do not recall them reported on other tryptophan metabolites. However, there are some significant concerns/comments I have to be addressed:

The study is n=15 of 22 patients on this phase I/II study. Please clarify if the 15 patients are the first 15 consecutive patients on the trial or if there are 7 patients in-between who refused to participate in the sub-study, since this is a potential introduction of bias in data collection for self reported PROs.

Only 3 patients developed severe neurotoxicity on the study, which makes it very difficult to identify biomarkers associated with severe neurotoxicity, and would be difficult to see any correlation in PROs and severe neurotoxicity. The most that can be said about these biomarkers is that they warrant validation in a larger cohort.

The CTCAE 4.02 neurotoxicity grading was used instead of the updated ASTCT system (Lee, BBMT 2018). Can the authors grade by ASTCT and are the results the same?

On page 9 line 203, reference 27 is cited for the cytokine data. Is this meant to be reference 26?

The 3HAA mean level is higher prior to CAR-T in those who develop neurotoxicity compared to those who don't, but there are some patients without neurotoxicity with higher levels, so I think it would be difficult to use as a sole biomarker, but perhaps useful in a biomarker panel.

Table 2 FSII row, column Day 90 mean, I believe is a typo (32.75).

Table 2, anxiety scores show that the mean is higher on day 28, and drops on day 90. Although the IQR is broad and the differences are not significant, it does make me wonder if anxiety peaked on day 28 and declined thereafter. Was the PRO questionnaire on day 28 and day 90 done before or after their disease-staging scan results were discussed with the patient? I'm assuming that the questionnaire was done prior, anxiety would be higher overall.

Reviewer #2 (Remarks to the Author):

This is an original study.

The N number is relatively low (15)

About assessing neurotoxicity, it would have been good to quantify neurofilament light (NfL) and/or GFAP.

Why in the figure 3 QA has only n=3, whereas the other metabolites have n=15? QA is the most neurotoxic compound of the kynurenine pathway.

If estimating sleep disturbance, I suggest to quantify melatonin and serotonin as well. If the KP is activated you should have a decrease in serotonin & melatonin levels.

Reviewer #3 (Remarks to the Author):

In this report, Knight et al. analyze patient reported outcome (PRO) data and blood kynurenine pathway metabolites in a small cohort of patients with relapsed refractory NHL treated with LV20.19 CAR T cells on a Phase I/IIb clinical trial. The trial results and serum cytokine analysis have been previously published (PMID: 33020647). In 15 patients who completed self-report surveys and had their blood analyzed for levels of tryptophan and 5 metabolites at baseline and at defined early timepoints (D14, D28, D90) post infusion, the authors find an association between change in patient reported depression from baseline to D14 and the rise in kynurenine in the blood. Among the 3 patients who develop grade 3-4 neurotoxicity they also find a significant correlation with elevated levels of 3HAA and 3HK compared to those with grade 0-1 neurotoxicity. They find no significant association with KYN, KA, or QA. The authors conclude that these findings suggest a role for kynurenine pathway metabolites in the development of neurotoxicity and that 3HAA and 3HK might be early indicators of neurotoxicity. They also find no significant change in patient reported QOL found in the first 90 days.

The results are intriguing and PRO studies are important for this patient population receiving a novel therapy but, given the small number of patients (15 total with only 3 developing severe neurotoxicity), they can only be considered preliminary.

All 3 of the patients who developed neurotoxicity also developed CRS, making it difficult to draw any conclusions about kynurenine metabolites and acute neurotoxicity specifically. These metabolite elevations may be an epiphenomenon related to CRS cytokine elevations, unrelated to the severe neurotoxicity symptomatology.

The authors indicate that the rationale for looking at kynurenine pathway metabolites is work by others showing an association between the metabolites and mood disorders. Acute severe neurotoxicity is not prominently characterized by mood changes, but rather severe global encephalopathy, aphasia, seizures, tremors, motor dysfunction, and rarely cerebral edema. Of the metabolites they study, the only one known to be associated with severe neurotoxic symptoms (ie seizures myoclonus) is QA, which was not found to be significantly elevated in their study.

If kynurenine metabolites are responsible for the neurotoxic process, wouldn't they expect to see

delayed side effects (depression) in the patients developing acute neurotoxicity who have ongoing elevations of 3HAA and 3HK at D28 and D90? A larger dataset seems critical here.

The mechanism by which the cytokines lead to 3HAA and 3HK elevations and their role in neurotoxicity and potential therapeutic strategies in the discussion are all largely speculative.

Finally, the association of the kynurenine pathway and CAR-associated acute neurotoxicity has been previously reported, including with CSF analysis, and should be referenced

DOI: 10.1158/2326-6074.CRICIMTEATIAACR18-A152

PMID: 29880584

Reviewer #4 (Remarks to the Author):

Secondary outcomes from a Phase I/Ib, first in human, bispecific, lentiviral, anti-CD20, anti-CD19 (LV20.19) CAR T cell clinical trial (NCT03019055), which accrued 22 patients are reported. Associations between patient-reported outcomes (PROs), tryptophan (TRP)-kynurenine (KYN) pathway metabolites, serum cytokines, and neurotoxicity at 15 days before CAR T cell infusion (baseline/apheresis) and 14, 28, and 90 days post-infusion were examined.

The primary analysis evaluated the impact of time on depression, anxiety, fatigue, sleep, and pain as assessed by various self-report measures among 15 patients with relapsed, refractory B cell non-Hodgkin's lymphoma or chronic lymphocytic leukemia/small lymphocytic lymphoma. Scores from the 20-item General Depression Subscale of the Inventory of Depression and Anxiety Symptoms (IDAS) were found to change significantly over time ($p = 0.03$). Relative to baseline, higher depression scores at Day 14 were maintained through Day 28 and observed to have decreased by Day 90. No significant change over time in fatigue, sleep, and pain was observed, and although data are not shown, the authors report no association between those PROs and neurotoxicity grade.

The secondary analysis evaluated the effect of time and neurotoxicity grade on TRP, KYN, putative neuroprotective metabolite, kynurenic acid (KA), and putative neurotoxic metabolites 3-hydroxykynurenine (3HK), 3-hydroxyanthralinic acid (3HAA), and quinolinic acid (QA). TRP, KYN, and KA concentrations significantly changed over time, with KYN concentrations showing the most change for patients with Grade 3/4 toxicity. Change in KYN was significantly correlated to change in scores from the General Depression Subscale of the IDAS from baseline to Day 14 ($p = 0.05$) and when pooled ($p = 0.002$). Baseline concentrations of KA, 3HAA, and 3HK were elevated among patients eventually experiencing neurotoxicity. Concentrations of 3HAA and 3HK were significantly elevated for patients with neurotoxicity Grade 3/4, relative to those with Grade 0/1. Interleukin 6 (IL-6) and 8 (IL-8) and tumor necrosis factor-alpha (TNF α) were negatively correlated with TRP and positively correlated with KYN and associated neurotoxic metabolites QA, 3HAA, and 3HK.

Strengths of the study include the assessment over time of PROs among patients accrued to a Phase I/Ib, first in human CAR T cell clinical trial, and the ability to correlate those measures with putative neuroprotective and neurotoxic metabolites and serum cytokines. Justification for the focus on the TRP-KYN pathway appears to hinge on evidence generated from studies of mood disorders. It is not apparent whether the focus on the pathway was post-hoc based on the finding that scores on the General Depression Subscale were found to change over time. It is not clear what to make of the

changes in scores on the General Depression Subscale. The authors characterize the cohort as one that reported "minimal depression, little (quality of life) disruption and minimal variability in self-reports of fatigue, sleep, and pain. However, as a clinician interested in the effects of CAR T cell therapy on symptoms associated with quality of life, I find the null findings cautiously encouraging, against the background of an 80% complete response to treatment.

Further, study results suggest support for the hypothesis that kynurenine metabolites, and 3HAA in particular, might be helpful as predictors of neurotoxicity onset following CAR T cell therapy. Although it need not be stated, the results are from a small Phase I/Ib trial, and additional study is definitively indicated. The authors might consider the introduction flow to improve the reader's ability to follow the scientific justification for focusing on the TRP-KYN pathway and measuring serum cytokines. For example, the authors might consider integrating the overview of cytokine release syndrome (lines 71-78) with the overview of the KYN pathway that begins on line 104 if the focus on this pathway was determined a priori. As it stands now, the justification that begins on line 104 is specific to mood disorders, yet the study also examined fatigue, pain, and depression. If the TRP-KYN pathway is implicated in the experience of the other PROs, it would be helpful to include that evidence. I recommend using terms that will resonate with clinical/medical readership (e.g., "biobehavioral" does not) and clarify whether the IDAS is used to assess clinical depression or symptoms associated with depressed mood. It would be helpful to reference clinically relevant thresholds for the IDAS and other measures used to assess PROs. I recommend revising the study's framing squarely on PROs rather than QOL since QOL does not appear to have been explicitly measured. Lastly, the authors might consider revising the manuscript to arrive at a succinct discussion of the correlative findings of this single-arm, unblinded, open-label trial.

Response to Reviewers

Reviewer #1:

1) “The study is n=15 of 22 patients on this phase I/II study. Please clarify if the 15 patients are the first 15 consecutive patients on the trial or if there are 7 patients in-between who refused to participate in the sub-study, since this is a potential introduction of bias in data collection for self reported PROs.”

- We now clarify in the first paragraph of the Methods section that the current study’s cohort of 15 participants is comprised of patients who enrolled *after* the IRB approved an amendment to collect PROs and biomarkers. This cohort consists of participants starting with patient 7 who were treated at the selected CAR-T dose.

2) “Only 3 patients developed severe neurotoxicity on the study, which makes it very difficult to identify biomarkers associated with severe neurotoxicity, and would be difficult see any correlation in PROs and severe neurotoxicity. The most that can be said about these biomarkers is that they warrant validation in a larger cohort.”

- We agree that these are preliminary data from an underpowered pilot study and that the incidence of neurotoxicity was low. We have clarified this point throughout the paper, including the first, fourth, and last paragraphs of the Discussion. The fourth paragraph of the Discussion further discusses the low incidence of neurotoxicity in particular. We note in the fourth and last Discussion paragraphs that these data are preliminary and warrant validation in a larger cohort.

3) “The CTCAE 4.02 neurotoxicity grading was used instead of the updated ASTCT system (Lee, BBMT 2018). Can the authors grade by ASTCT and are the results the same?”

- In the primary Shah et al paper the authors retrospectively regraded neurotoxicity utilizing the updated ASTCT system and based on that there was no change in neurotoxicity grade (Supplemental Appendix Table 3). This is now noted in the *Neurotoxicity* section of the Methods.

4) “On page 9 line 203, reference 27 is cited for the cytokine data. Is this meant to be reference 26?”

- Yes; we have corrected this to be reference 26 and not 27. Thank you for pointing this out.

5) “The 3HAA mean level is higher prior to CAR-T in those who develop neurotoxicity compared to those who don’t, but there are some patients without neurotoxicity with higher levels, so I think it would be difficult to use as a sole biomarker, but perhaps useful in a biomarker panel.”

- We agree and have clarified our discussion of the utility of 3HAA as potentially useful in a biomarker panel, and not as a sole biomarker (first 2 paragraphs of Discussion).

6) “Table 2 FSII row, column Day 90 mean, I believe is a typo (32.75).”

- Yes; we have corrected this misplaced decimal.

7) “Table 2, anxiety scores show that the mean is higher on day 28, and drops on day 90. Although the IQR is broad and the differences are not significant, it does make me wonder if anxiety peaked on day 28 and declined thereafter. Was the PRO questionnaire on day 28 and day 90 done before or after their disease-staging scan results were discussed with the patient? I’m assuming that the questionnaire was done prior, anxiety would be higher overall.”

- The reviewer asks a great and pertinent question. Unfortunately, given the clinical care schedule and automatic release of records, including disease-staging scans, direct to the patient through the electronic medical record, these data are not ascertainable. This is an important point to consider and potentially collect specific data on in future studies. This is now noted in the limitations section of the Discussion section.

Reviewer #2:

1) “The N number is relatively low (15).”

- As per our response to Reviewer #1, comment 2, we now note more clearly throughout the manuscript that these are preliminary data from a pilot study and warrant further validation in a larger cohort.

2) “About assessing neurotoxicity, it would have been good to quantify neurofilament light (NfL) and/or GFAP.”

- We now include discussion of this limitation and future direction in the third to last paragraph of the manuscript.

3) “Why in the figure 3 QA has only n=3, whereas the other metabolites have n=15? QA is the most neurotoxic compound of the kynurenine pathway.”

- The assay utilized was meant to analyze all kynurenine metabolites and was therefore not optimized to measure QA in particular; subsequently there were only values for 6 participants at baseline and 4 at D90. This is now indicated in the “Measurement of tryptophan and metabolites” portion of the Methods section.

4) “If estimating sleep disturbance, I suggest to quantify melatonin and serotonin as well. If the KP is activated you should have a decrease in serotonin & melatonin levels.”

- This is a potentially very interesting line of inquiry. Unfortunately these neurochemicals were not measured in our assay, nor do we have the capability now based on the samples collected. Assessment of melatonin and serotonin can be considered in future studies.

Reviewer #3:

1) “The results are intriguing and PRO studies are important for this patient population receiving a novel therapy but, given the small number of patients (15 total with only 3 developing severe neurotoxicity), they can only be considered preliminary.”

- As per our responses above to Reviewer #1, comment 2 and Reviewer #2, comment 1, we now note more clearly throughout the manuscript that these are preliminary data from a pilot study and warrant further validation in a larger cohort.

2) “All 3 of the patients who developed neurotoxicity also developed CRS, making it difficult to draw any conclusions about kynurenine metabolites and acute neurotoxicity specifically. These metabolite elevations may be an epiphenomenon related to CRS cytokine elevations, unrelated to the severe neurotoxicity symptomatology.”

- We agree it is important to continue to differentiate the underlying pathophysiological differences and similarities between CRS and neurotoxicity. As the reviewer has noted above, the current sample size is small and thus precludes our ability to derive substantial meaning from further subanalyses. However, we agree this is important to consider in future studies of larger cohorts. We now note this in the limitations section of the Discussion.

3) “The authors indicate that the rationale for looking at kynurenine pathway metabolites is work by others showing an association between the metabolites and mood disorders. Acute severe neurotoxicity is not prominently characterized by mood changes, but rather severe global encephalopathy, aphasia, seizures, tremors, motor dysfunction, and rarely cerebral edema. Of the metabolites they study, the only one known to be associated with severe neurotoxic symptoms (ie seizures myoclonus) is QA, which was not found to be significantly elevated in their study.”

- We now state in the 2nd paragraph of the Introduction that CAR T neurotoxicity has not yet been established as a precursor to subsequent changes in mood and other related symptoms. We also reference the association between QA and neurotoxicity as described by Santomasso et al. in the 5th paragraph of the Introduction and discuss our disparate findings in the second paragraph of the Discussion section. Finally, as also suggested by Reviewer #4 in comment 4, we have reworked the Introduction to clarify the rationale for our investigation of the kynurenine pathway, mood and related symptomatology, and neurotoxicity.

4) “If kynurenine metabolites are responsible for the neurotoxic process, wouldn’t they expect to see delayed side effects (depression) in the patients developing acute neurotoxicity who have ongoing elevations of 3HAA and 3HK at D28 and D90? A larger dataset seems critical here.”

- We strongly agree with the reviewer’s important point that delayed side effects such as depression may exist in individuals who develop neurotoxicity, particularly if they have ongoing elevations of 3HAA and 3HK (and/or other neurotoxic biomarkers). It is unknown how long this might take to develop, and could reasonably be delayed beyond even D90. As noted in our response to this reviewer’s comment 1, we now note more clearly throughout the manuscript that these are preliminary data from a pilot study and

warrant further validation in a larger cohort, particularly where more patients develop neurotoxicity and are able to be followed for a longer duration.

5) “The mechanism by which the cytokines lead to 3HAA and 3HK elevations and their role in neurotoxicity and potential therapeutic strategies in the discussion are all largely speculative.”

- We have reworded the language and presentational style around the cytokines/3HAA/3HK/neurotoxicity relationship as well as potential therapeutic strategies to be modestly suggestive as a result of our preliminary pilot data, but not definitive (4th paragraph of Discussion).

6) “Finally, the association of the kynurenine pathway and CAR-associated acute neurotoxicity has been previously reported, including with CSF analysis, and should be referenced.”

- We now reference Santomasso’s finding in the 5th paragraph of the Introduction.

Reviewer #4:

1) “Strengths of the study include the assessment over time of PROs among patients accrued to a Phase I/Ib, first in human CAR T cell clinical trial, and the ability to correlate those measures with putative neuroprotective and neurotoxic metabolites and serum cytokines. Justification for the focus on the TRP-KYN pathway appears to hinge on evidence generated from studies of mood disorders. It is not apparent whether the focus on the pathway was post-hoc based on the finding that scores on the General Depression Subscale were found to change over time. It is not clear what to make of the changes in scores on the General Depression Subscale.”

- We now clarify our presentation and focus on mood and patient-reported symptomatology throughout the Introduction and as indicated above in response to Reviewer #3, comment 3, clarifying that assessment of the TRP-KYN pathway was indeed an *a priori* planned assessment. We agree that these data – including changes in the depression scores – are preliminary and warrant validation in a larger cohort (as noted and responded to in prior responses to reviewers as above).

2) “The authors characterize the cohort as one that reported “minimal depression, little (quality of life) disruption and minimal variability in self-reports of fatigue, sleep, and pain. However, as a clinician interested in the effects of CAR T cell therapy on symptoms associated with quality of life, I find the null findings cautiously encouraging, against the background of an 80% complete response to treatment.”

- We agree these findings are cautiously optimistic, though we will need larger prospective trials to understand this more definitively (as detailed to Reviewers #1-3 above in comments 2, 1, and 1, respectively).

3) “Further, study results suggest support for the hypothesis that kynurenine metabolites, and 3HAA in particular, might be helpful as predictors of neurotoxicity onset following CAR T cell therapy. Although it need not be stated, the results are from a small Phase I/Ib trial, and additional study is definitively indicated.”

- Again, we agree and have emphasized the pilot nature of our study throughout the manuscript as detailed in prior responses to reviewers and noted in our response to comment 2 above.

4) “The authors might consider the introduction flow to improve the reader's ability to follow the scientific justification for focusing on the TRP-KYN pathway and measuring serum cytokines. For example, the authors might consider integrating the overview of cytokine release syndrome (lines 71-78) with the overview of the KYN pathway that begins on line 104 if the focus on this pathway was determined a priori. As it stands now, the justification that begins on line 104 is specific to mood disorders, yet the study also examined fatigue, pain, and depression. If the TRP-KYN pathway is implicated in the experience of the other PROs, it would be helpful to include that evidence.”

- We thank the reviewer for this suggestion to improve the flow of the introduction and scientific justification, as focusing on the TRP-KYN pathway was indeed determined *a priori*. We now introduce the TRP-KYN pathway earlier in the Introduction and limit discussion of CRS so as better to retain clinical focus on neurotoxicity.
- We now clarify that we are assessing mood *and related symptomatology* and that the TRP-KYN pathway is additionally associated with the related symptoms of fatigue, sleep, and pain (new references 6, 7, and 8, respectively) in the 2nd paragraph of the Introduction.

5) "I recommend using terms that will resonate with clinical/medical readership (e.g., "biobehavioral" does not) and clarify whether the IDAS is used to assess clinical depression or symptoms associated with depressed mood."

- We have replaced 'biobehavioral' in the 4th paragraph of the Introduction and clarified in the *Patient-reported outcomes* section of the Methods that the IDAS is used to assess the dimensionality of depressive symptoms.

6) "It would be helpful to reference clinically relevant thresholds for the IDAS and other measures used to assess PROs."

- We have added in more detail on PRO metrics in the PRO section of the Methods.

7) "I recommend revising the study's framing squarely on PROs rather than QOL since QOL does not appear to have been explicitly measured."

- We have replaced QOL with PRO terminology throughout when referencing current study findings.

8) "Lastly, the authors might consider revising the manuscript to arrive at a succinct discussion of the correlative findings of this single-arm, unblinded, open-label trial."

- We now include a succinct summary that these are hypothesis-generating correlative findings from a single-arm, unblinded, open-label trial of bispecific LV20.19 CAR T cell therapy upon which to guide future investigation as the first sentence of the Discussion section.

REVIEWERS' COMMENTS:

Reviewer #1 (Remarks to the Author):

The authors have adequately addressed my concerns in this revised manuscript, and I have no further comments.

Reviewer #3 (Remarks to the Author):

The authors have tempered their claims and appropriately acknowledged the significant limitations of the study, producing an overall better report. All of my comments have been adequately addressed.

Minor

Beyond the very small number of patients and samples analyzed, the authors might want to acknowledge in the discussion that blood perturbations of cytokine and kynurenine pathway metabolites may not be reflective of CSF/central nervous system changes. This may be yet another reason why their data did not demonstrate any significant associations between QA and neurotoxicity.

For the patient's who developed neurotoxicity, the exact symptomatology, timing, and duration of symptoms post infusion (ie day 5-8) should be included in the manuscript, perhaps in a supplemental table.

Reviewer #4 (Remarks to the Author):

Thank you for thoughtfully addressing concerns raised by Reviewer 4. I have no additional substantive concerns.

Response to Reviewers

Reviewer #1:

“The authors have adequately addressed my concerns in this revised manuscript, and I have no further comments.”

- Thank you for your previous comments and the opportunity to address them.

Reviewer #3:

“The authors have tempered their claims and appropriately acknowledged the significant limitations of the study, producing an overall better report. All of my comments have been adequately addressed.”

- We are glad to have adequately addressed your comments and agree this has resulted in a better report.

1. “Beyond the very small number of patients and samples analyzed, the authors might want to acknowledge in the discussion that blood perturbations of cytokine and kynurenine pathway metabolites may not be reflective of CSF/central nervous system changes. This may be yet another reason why their data did not demonstrate any significant associations between QA and neurotoxicity.”

- We now acknowledge this additional explanation for our findings in the Discussion on p. 16.

2. “For the patients who developed neurotoxicity, the exact symptomatology, timing, and duration of symptoms post infusion (ie day 5-8) should be included in the manuscript, perhaps in a supplemental table.”

- We now include a Supplemental Table 1 with the requested data, referenced in the manuscript on p. 12.

Reviewer #4:

“Thank you for thoughtfully addressing concerns raised by Reviewer 4. I have no additional substantive concerns.”

- We thank the reviewer for their initial comments and are pleased they were received well.